# A New Top-Down Governance Approach to Community Gardens: A Case Study of the "*We Garden*" Community Experiment in Shenzhen, China

**Xunyu Zhang [1], Dongxu Pan [2], Kapo Wong [3] and Yuanzhi Zhang [4,5,*]**

[1] The Nature Conservancy, Shenzhen 518053, China; xunyu.zhang@tnc.org
[2] Shenzhen Development Research Center for Real Estate and Urban Construction, Shenzhen 518028, China; pandongxuchn@163.com
[3] School of Nursing, The Hong Kong Polytechnic University, Hong Kong 999066; portia.wong@polyu.edu.hk
[4] Center for Housing Innovations, Institute of Asia-Pacific Studies, Faculty of Social Science, Chinese University of Hong Kong, Hong Kong 999077
[5] Faculty of Marine Sciences, Nanjing University of Information Science and Technology, Nanjing 210044, China
[*] Correspondence: yuanzhizhang@cuhk.edu.hk

**Abstract:** Over the past few decades, development in China (including Shenzhen) has been led by the State, meaning that the government has been responsible for major decisions in urban construction and management. However, the current enormous contradiction between people's demand for livability and Shenzhen's unequal and inadequate urban development means that leaving all the administrative work to the government alone has become unsustainable. Since 2020, Shenzhen has introduced a new urban management approach called "*We Garden*", in which the government supports public participation aimed to transform idle public lands into green spaces in the form of community gardens. Because this ongoing but novel community garden experiment is a recent development in China, literature investigating the phenomenon context, especially the associated motivations and governance structure, remains scarce. This paper aims to clarify the governance structure and operation mechanism of the Shenzhen community garden program through all stages: from planning and design through construction or implementation to management. Fieldwork with active participation, direct observation, and semi-structured, qualitative interviews as participant in a nonprofit organization revealed that the Shenzhen experiment was driven by urban environmental public governance rather than individual needs. The community garden development approach is a new top-down governance structure that expands on existing governance types in the literature, while emphasizing the key role that nonprofit organizations play in the process. Therefore, this new governance approach expands beyond the environmental improvement of urban communities, serving as a new mechanism for sustainable public participation in urban environmental protection.

**Keywords:** community garden; governance; top-down; Shenzhen; nonprofit organization

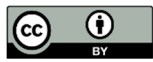

## 1. Introduction

For decades, the development process of China has been State-led, emphasizing the government's role in making major decisions about urban construction and management [1]. However, China's continuing economic growth and the improvement of urbanization (with the urbanization rate reaching a new level at 63.89%) [2]) have led social development to enter a new stage where citizens, no longer satisfied with basics such as food and clothing, have also developed high expectations regarding the quality of life and their

living environment. Therefore, a large contradiction has emerged between people's demand for livability and the inequality and inadequacy that have characterized urban development. In essence, this contradiction represents a conflict between population and resources that has become prominent in Shenzhen and is at the forefront of China's economic development. Overs the course of only 40 years, Shenzhen, China's first Special Economic Zone, has become a megacity, exhibiting an explosive growth from 0.31 million to 17.56 million permanent residents, a 55-fold increase within the span of four decades. Shenzhen, which is the only city in China with an urbanization rate of 100% [3], retains about 50% of the ecological control line (In other words, the city has set aside 50% of its land to function as green lungs), severely limiting the amount of developable urban land. However, in a high-density urban environment, the green and public urban space left to the public falls far short. In the meantime, migrant young people who have become new citizens have put forward higher requirements for the quality of the living environment. The rising grassroots demands have created a situation where guidance by the government alone has become unsustainable. Therefore, in 2020, following the grassroots governance concept of "We are the cities we make" that was initiated at the national level [4], the Shenzhen Municipal Government proposed a new urban management approach called *"We Garden"*. This emerging community trend in Shenzhen is transforming idle public land into green space in the form of community gardens. The government formulated the scheme with associated financial support, encouraging public participation. The local government has announced plans to build 120 community gardens in Shenzhen per year to improve the greening rate in high-density residential areas [5]. The launch of this new scheme, along with official encouragement, has led different stakeholders from government institutions, nonprofit organizations, schools, communities, and professional agencies (e.g., design and gardening companies) to come together to implement community gardens on the ground, transforming the project into a process of co-construction and governance.

In this context, developing community gardens in Shenzhen has gone beyond the basic concept of a planting space and social space in the community and is now poised to solve the contradiction between the city's future environmental development and the ubiquitous shortage of land and financial pressure, while simultaneously encouraging grassroots autonomy. Once one community has successfully implemented the idea, the process can be replicated and promoted at the municipal level at a low cost. In terms of the process of community garden planning, construction, and management, it was also found that its motivation and governance structure differed from typical patterns illustrated by the study.

Scant literature is available regarding this type of ongoing, novel community garden experiment in the Chinese context. Accordingly, this paper aims to clarify the governance structure and operation mechanism of the Shenzhen community garden program through all stages: from planning and design through construction or implementation to management. Furthermore, this investigation delved into the new mechanism of public participation in urban environmental protection in the Chinese context.

The study was based on the following research questions: (1) What is a diversity in participants/actors and their action styles? (2) How is the new community garden scheme carried out in local contexts, and what were the results? (3) How can the Shenzhen case help scholars better theorize the governance structure in community gardens and beyond the city level?

This discussion addresses these questions and identifies the relevant gaps that emerged in the research process through an initial review of the research status, motivations for community gardens, and types of governance structure in this context. Then, as a reflection of nonprofit organizations and deeply involved in the development of community gardens, we describe the study's research methods. Next, the paper will present

the findings from the interviews, along with details of the process and results of the development of a particular community garden. Lastly, the focus of the discussion of the findings will center around governance structure.

By comparing the governance structure of Shenzhen with other community gardens worldwide, we believe that the model in Shenzhen (described here as top-down with public engagement driven by nonprofit organizations) complements the governance structure of community gardens while also representing an innovative mechanism to promote public participation in urban environmental protection. This model, which will allow members of society to fully participate in the construction of green cities in the future, is innovative and reproducible from an international perspective.

## 2. Literature Review

A community garden—distinguished from a private garden [6]—refers to open spaces managed and operated by local community members, which allow people to work together to grow herbs, fruit, vegetables, or flowers [7–10]. It is widely acknowledged that the contemporary community garden originated in the United States in the 1970s [11,12], although some researchers have asserted that the idea originated from the two World Wars [13,14]; in any event, all conclusions point to the appearance of this phenomenon in terms of people who decided to grow food to contribute toward self-sufficiency.

To date, the number and diversity of community gardens have significantly increased. There are several streams of research on community gardens. The main stream of research has featured the social sciences, including but not limited to planning, health, geography and sociology, and education, covering topics such as social capital [15–18], health and well-being outcomes [13,19–21], and community engagement and development [7,22,23].

In our review of the previous research, we focused on the literature on the motivations, as well as governance approaches/organizational structures of community garden projects. Among the insights that emerged, we found that while community gardens can serve as an alternative food/medicine to provide economic benefit and supply healthy food and supplies, at the same time, they can also provide open spaces for social activities or recreation [24,25]. Multiple studies have reported similar motivations for building community gardens, such as consuming fresh foods, making or saving money by eating from the garden or selling produce, social cohesion, and improving health. Other identified motivations included education, enjoying nature, and enhancing environmental sustainability [26].

Another discovery was that community gardens usually start from bottom-up efforts by a grassroots community, individuals, or groups [13,17,27]. In addition to the bottom-up approach, a top-down structure organized by the public sector (e.g., municipal agencies) provide an essential complement to community garden development [28]. After comparing community gardens between fall 2011 and fall 2016, Rees & Melix (2019) found a disappearance of gardens established from the top-down and an increase in grassroots neighborhood gardens from the bottom-up, which they thought is a more pragmatic approach [29]. The bottom-up approach encourages more participation and commitment from people [30], which might improve the likelihood of a garden's longevity [31,32]. However, Rosol (2005) and Baker (2004) argued the top-down approach was a good way to ensure the effective management of gardens [33,34]. Specifically, McGlone et al. (1999) subdivided community gardens into five different governance structures under the two mentioned basic modes: top-down, top-down with community help (gardens planned, established, or managed by paid professionals with community involvement), bottom-up, bottom-up with professional help, and bottom-up with informal help [35]. Fox-Kämper et al. (2017) later expanded this theory by adding a new category called bottom-up with political and/or administrative support (PAS), which refers to using a bottom-up approach with some external support, mostly involving local government agencies that provide land, funding, and/or expertise [36].

A comparison of motivations and governance approach reveals that the ongoing Shenzhen community garden experiment differed from the previously described definitions in terms of motivation and governance structure in the Chinese context. The following points present four limitations of the prior studies and how we addressed them:

(1) Research background. Overall, few studies have taken place in non-English speaking countries [26]. In China, community gardens involving public participation in construction and management have only emerged in this decade [37]. Thus, in China (mainly in Shanghai) [38–40] and especially in Shenzhen, very little research has been conducted, and understanding is lacking in this area, especially in terms of knowledge about how community gardens work in a Chinese context. As members involved in communication and coordination with actors, we delved deeply into the community garden in Nanshan district, Shenzhen, which can be seen as representative of China's evolving local socialist market economy in pursuit of sustainable development [41].

(2) Governance approach. The governance structure of the ongoing Shenzhen community garden development appears to be top-down with community help or PAS literally. After conducting a content analysis of articles with information on governance structures, several cases of community gardens using these two models are searched and summarized as follows: As for top-down with community help, first, most cases were involved in paid professionals from political and administrator support. For example, the Philadelphia Urban Gardening Program in America was sponsored by university cooperative extension service with technical assistance and supported by the Philadelphia Horticultural Society with materials [42], 'Dig In' community garden projects in England, Australia (same as the British case), Cape Town, and Shakashead community gardens in South Africa were all supported from a professional organization with a dedicated trust or funding [17,29,30,43]. Second, several community gardens initiated a top-down approach in the planning and implementation phases, although they underwent the transition from top-down to bottom-up during the management phase [9,10,17,44]. As for PAS, despite being a bottom-up initiative, projects have received significant support such as financial or advisory support, donations of materials, free water supply. Some received support during the construction and implementation phase, which is a common form in the planning and design phase [38,45–48]. After comparison, the governance structure of Shenzhen still has unique qualities. More information about the ongoing Shenzhen community garden governance structure is needed to promote a complete understanding, reflecting Guitart et al.'s (2012) suggestion that "future studies should shed light more on how participants garden" [26]. In focusing on the governance of the community garden, as well as the construction process, a deep understanding of the processes around participation in the community garden was gained.

(3) The role of each participant (especially nonprofit organizations). In general, the success of a community garden depends on the degree of its internal organization [17]. For example, in the case of a temporary post-earthquake community garden in central Christchurch, New Zealand, a local organization—Greening the Rubble (GtR)—acted as a coordinator that created new, linked social capital that was vital to the success of the project [48]. Similarly, even though the success of the community garden project in Shenzhen has also been closely related to nonprofit organizations, only a few studies have explored the active role of nonprofit organizations in the development of community gardens. This topic could be explored because we were involved in the entire process of community garden construction and management.

(4) Motivations. In the international or traditional sense, most community gardens are initiated on the community scale to meet individual's needs, though some are directly initiated by project managers and institutions [26]. From the perspective of future urban environmental development, few governments have directly issued community garden plans. That is to say, the motivation behind the ongoing Shenzhen

community garden has changed from meeting the material, spiritual, and cultural needs of individuals to an urban management approach in which the government issued policies and encouraged public participation to realize the sustainable development of the city. Through the interview with government officials, the Shenzhen community garden was discussed in terms of giving a new mission.

## 3. Material and Methods

The section comprises three parts. The first introduces the overall situation of community gardens in Nanshan District of Shenzhen and describes the selected community garden program. The second explains the six groups of participants in this experiment and their characteristics. The third discusses the data collection protocol, which unfolds participatory observations and semi-structured, qualitative interviews.

### 3.1. Case Selected

Nanshan District government announced that 20 community gardens will be built through public participation per year from 2020 to 2025, which means that by the end of 2025, the number of community gardens will be 100. The number of gardens, which is also the work KPI of the government, is larger than that in other districts due to the financial income level of the district.

From 2020 to 2021, as a member of the Nature Conservation (TNC, one of the world's famous Non-Government Organizations), the author participated in the whole process, from planning to completion, of 12 community garden projects in the Nanshan District. During this process, TNC, as an environmental protection organization, was responsible for providing nature-based solutions and promoting public participation. Each project took an average of nearly 3 months, and multiple projects were carried out in parallel. It is found that the governance approach would gradually tend to be the same in general. Therefore, we will select one typical case—*Pen Garden* (The names mentioned in this paper are pseudonyms)—for discussion.

*Pen Garden* is in an idle public green space on the roadside of Changhai Community in Nanshan District, covering a total area of about 800 square meters, and the garden renovation area is about 400 square meters. The plot has several characteristics: (1) The plot is owned by the government, avoiding insecurities such as other American community gardens being withdrawn after a certain duration [26]; (2) It was originally a public green space furnished with garden pavements and seats, but due to the lack of management, the garden is overgrown with weeds, making it difficult for people to enter, thus becoming a negative green space of urban waste (Figure 1); (3) It is located next to the traffic artery of the community and close to commercial residential areas, schools, and shops. Thus, the plot has a large flow of people every day, which can be utilized by surrounding residential residents, schools, enterprises, and even passers-by. Considering the existing issues and the promising greater social benefits, it was determined and supported by the government.

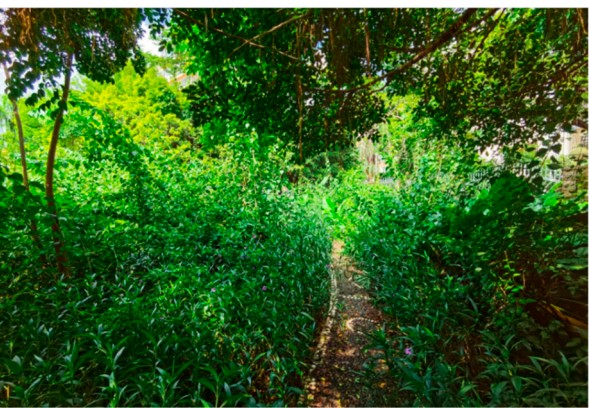

**Figure 1.** *Pen Garden* before the transformation. (Source: ZHOU Yu, 2020).

### 3.2. Participants Characteristics

In the *"We Garden"* project, all participants could be mainly divided into four categories: The first is the officials of government institutions. The effect of the community garden is their work KPI, but as a public participation project, they need to make a balance between their work standards and the needs of residents. The second refers to nonprofit organizations, including: (1) Community organizations, including foundations and Party member service centers rooted in the community. They have community influence and appeal and are essential to fostering public participation. (2) Non-Governmental Organizations that are not rooted in the community but have more technologies or resources help the community link more social capital. The third category is landscape designers from design companies. Different from the previous projects who are commissioned to design, this time they served as technical representatives to support community gardening and help the residents without any experience realize their design concepts in garden construction. The fourth category is communities, who studied, lived, or worked within about a walking distance of 500 m of the garden (in the *Pen Garden* project, mainly students). They have the greatest right to enjoy the garden and have the obligation to maintain it. The other small part is volunteers from outside the community who participate in the project.

### 3.3. Data Collection Methods

The methods of data collection included participant observations and semi-structured and qualitative interviews. As members of nonprofit organizations who were deeply involved in the development of community gardens, we collected information during site gardening and via meetings, impressively exploring how the various actors in this project performed through active participation and direct observations.

In addition, semi-structured, qualitative interviews were conducted between October 2020 and January 2021 among ten of the *Pen Garden* project actors (Table 1). They were recorded and transcribed. Quotations from Chinese interviews were translated into English by the authors. The interview mainly includes the following questions: (1) Their roles in the development of community garden development, and (2) Evaluations before and after participation.

**Table 1.** Profile of interviewees.

| Interviewee | Position | Organization | Sector |
|---|---|---|---|
| Interviewee A | Deputy director | Urban Management Bureau of Shenzhen Municipality | Government |
| Interviewee B | Director | Urban Management Bureau of Shenzhen Municipality Nanshan District | Government |
| Interviewee C | Officer | Urban Management Bureau of Shenzhen Municipality Nanshan District | Government |
| Interviewee D | Officer | Shekou Community Foundation | Nonprofit Organization |
| Interviewee E | Coordinator | Shekou Community Foundation | Nonprofit Organization |
| Interviewee E | Landscape Designer | Local council | Designing Enterprise |
| Interviewee G | Landscape Designer | Local council | Designing Enterprise |
| Interviewee H | Teacher | Community School | Public institute |
| Interviewee I | Student | Community School | Public institute |
| Interviewee J | Student | Community School | Public institute |

## 4. Results

In addition to the governments, each actor involved in "*We Garden"* played a particular role at different stages (including planning, pre-research, co-design, co-building/co-construction, and maintenance phase) in the community garden development. Their roles and action styles were listed in Figure 2. Among them, nonprofit organizations were involved throughout the process and were essential in promoting public engagement, as well as the success of the project. In the first four sections below, we took *Pen Garden* as an example to illustrate how nonprofit organizations cooperate with the government, communities, and professionals in the whole phase. In the last section, we listed the representative feedback from the government director, designer, and student on participating in the transformation process of *Pen Garden* with the help of nonprofit organizations.

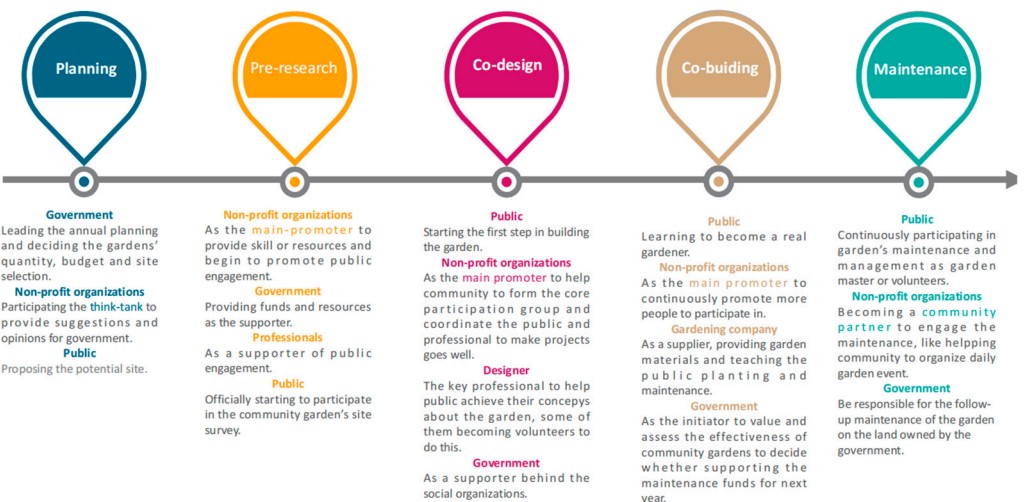

**Figure 2.** Roles and action styles of each actor in the whole phases of *We Garden* community garden development. (Source: Author).

### 4.1. The Government Involved Nonprofit Organizations in the Planning Phase

In the planning phase, the Nanshan district government took an open and innovative approach, inviting different nonprofit organizations to take part in a think tank to provide professional suggestions. Examples include TNC, which focused on urban environment conservation, and the Community Foundation, whose efforts centered around community service and governance. In the meantime, nonprofit organizations also facilitated government understanding of people's needs and provided feedback for administrative decision making. For example, when implementing the "*We Garden"* scheme, the government initially approached the community garden project as a topic for consideration; that is, it issued administrative instructions to start the site-selection process, and requested that the sub-district offices perform the preliminary investigation and make decision concerning site selection. Although the site may be located in the community, this type of compulsory approach may eventually discourage participation if it leads to situating the community garden at an inconvenient distance from the residents' activity area or causes a lack of public understanding. Therefore, the Community Foundation advised the government to adopt an open approach by encouraging the public to offer input about community garden site selections. This technique facilitated the collection of the real needs of the public and is a prerequisite for promoting public participation. The government adopted this advice, released the news of the opportunity to the public through social media, and encouraged them to submit suggestions concerning idle space in the community that they wanted to transform into a community garden. If a location qualified, the government would provide financial support according to the garden's location, area, and situation. *Pen Garden* is an example of such a garden recommended by the public. As the

garden is on public land owned by the government, it can generate greater social benefits after evaluation, and for that reason, it has received government support.

*4.2. Nonprofit Organizations Helped the Community Promote Public Engagement*

In the next stage, the public needed to be involved in the co-design and co-construction of the community garden. This part of the process represents an area where nonprofit organizations could help with various skills and resources. In this case, *Pen Garden* had been neglected by the surrounding residents for a long time, making public participation difficult to attract. Therefore, the Community Foundation and TNC cooperated and began their efforts at a school next to the site. In cooperation with the schools, they represented the community garden project as a social practice course for students and organized an environmental education workshop, inviting a professional team of animal and plant experts and garden landscape designers to lead students in carrying out site research (Table 2, Figure 3). This workshop was recognized by the government and the school. Teachers and students gradually became interested in community garden construction and became the main participants in this project. In addition, the Community Foundation recruited volunteers to participate via social media; nevertheless, compared with the stable groups in the school, most of the participants attracted by this temporary recruitment method were one-time contributors whose involvement was unsustainable.

**Table 2.** The sample site plant survey questionnaire.

| Garden: *Pen Garden* | | Recorder: XXXX | | | Date: XXXX | |
|---|---|---|---|---|---|---|
| No. | Species Name | Density Class | Number | Coverage | Growth | Other |
| 1 | | | | | | |
| 2 | | | | | | |
| … | | | | | | |

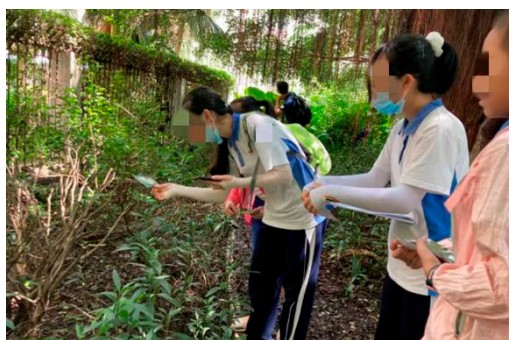 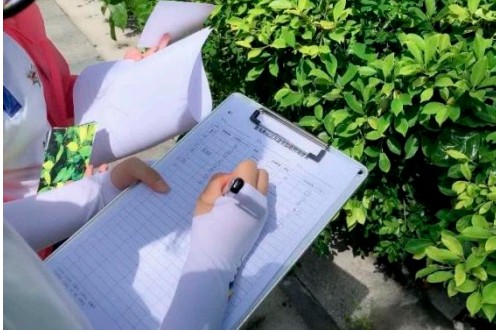

**Figure 3.** The students conducting the site survey. (Source: Action, 2020).

*4.3. Nonprofit Organizations Helped the Community to Reach Out to More Professional Resources*

Only a few communities have experienced gardeners, and these individuals usually become the leaders of community gardens and organize their communities to carry out garden construction and maintenance. However, most of the participants had no gardening experience and consequently needed training and guidance. Under the circumstances, nonprofit organizations were able to use the resources and networks of social organizations to help them quickly find appropriate professional resources. In the *We Garden* projects, the government adopted TNC's proposal and held a community garden design competition with local foundations, design competition platforms, and other institutions to solicit good works. The winner received government funds as the competition prize

through the local foundation. Some designers also participated as volunteers, working with the public to realize their design concepts. At the same time, nonprofit organizations also organized workshops and meetings to facilitate public consultation with experts and continuously promoted increasing participation.

In the *Pen Garden* project, after completing the ecological background investigation, TNC found that the original biodiversity of this place was relatively rich, making the location suitable to be transformed into a habitat garden, meaning that the focus was on protecting the area's original natural resources through low-impact design. Accordingly, TNC asked for a suitable designer for the community and organized a workshop. The workshop process included: (1) Discussion and analysis of the site survey results to determine the transformation intention characterizing the garden renovation; (2) Sketching and modeling the community garden; (3) Group sharing and exchanging ideas to determine the final plan. Twenty students, divided into five groups, participated in the whole process. Some of them proposed preserving shade plants on the site, while others come up with the idea of creating a bird habitat. Another topic of focus involved older adults' need for a restful oasis. Finally, these ideas were integrated into the garden design (Figure 4).

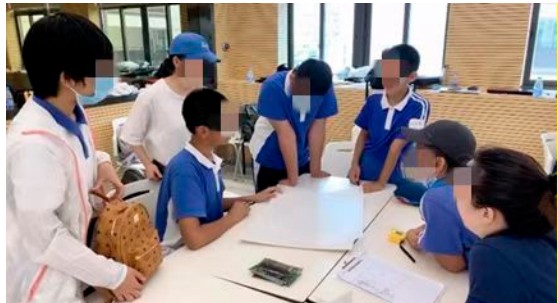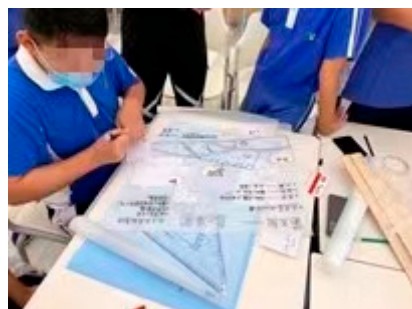

**Figure 4.** The students in the design workshop. (Source: Action, 2020).

The cooperation described here continued into the garden construction process, gradually promoting increasing intensive public participation, which helped the community to establish a volunteer group to participate in the follow-up maintenance and management. As a result, *Pen Garden* retained the original vegetation of the site, complemented local plants, set up bird "rooms" and bathing basins with plants, created bird habitats, and added seats and sand pools for children's play. This location now features an elder-friendly, child-friendly, and rich-in-biodiversity garden, providing natural well-being for the people who live or work in proximity to the site (Figures 5–7).

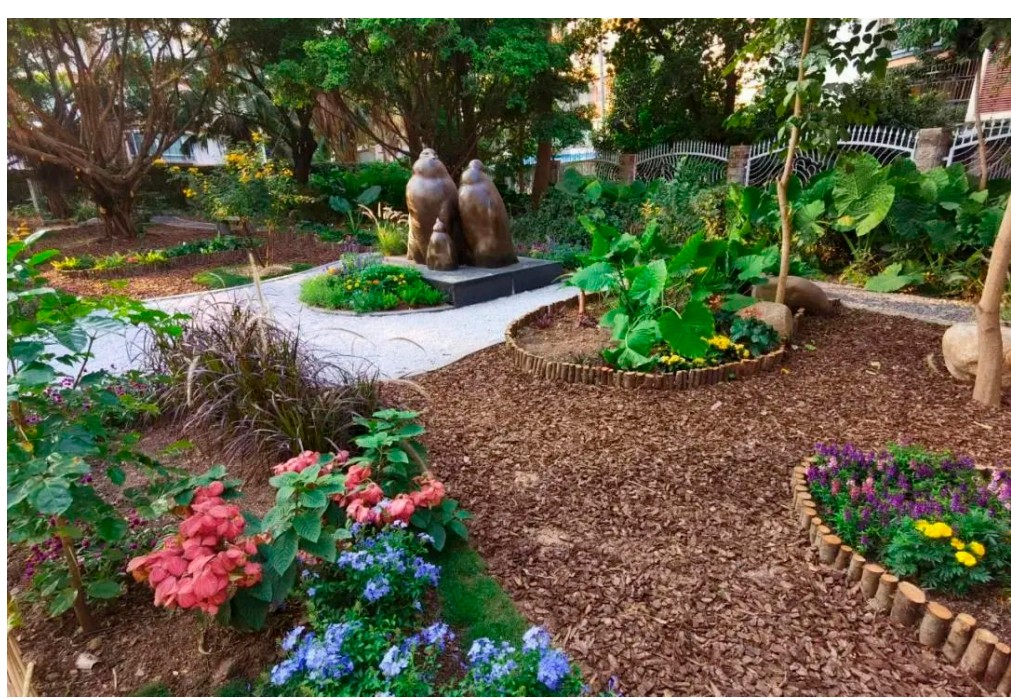

**Figure 5.** *Pen Garden* after transformation. (Source: ZHOU Yu, 2020).

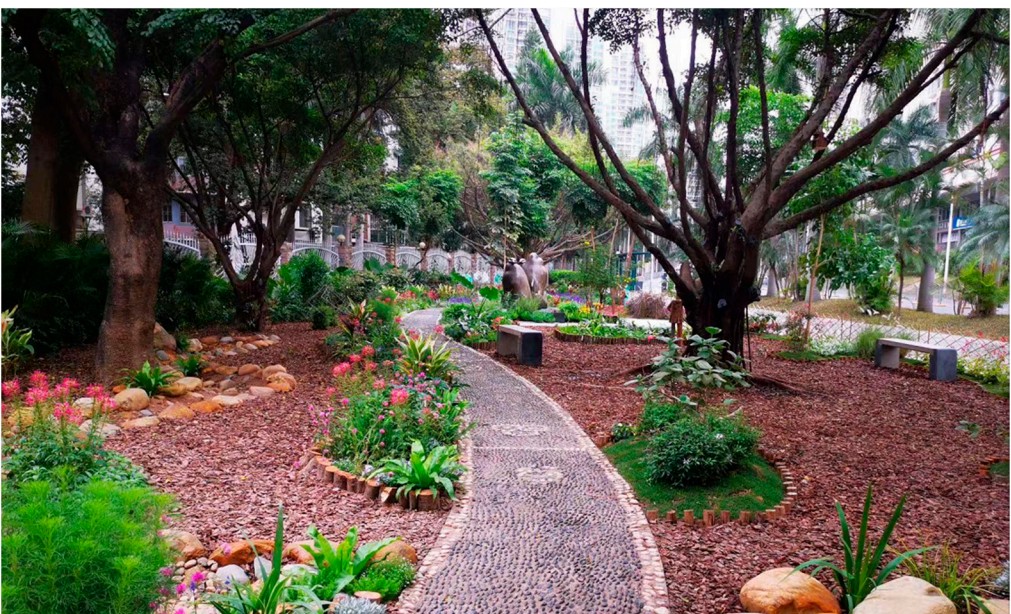

**Figure 6.** *Pen Garden* after transformation. (Source: CHEN Feng, 2020).

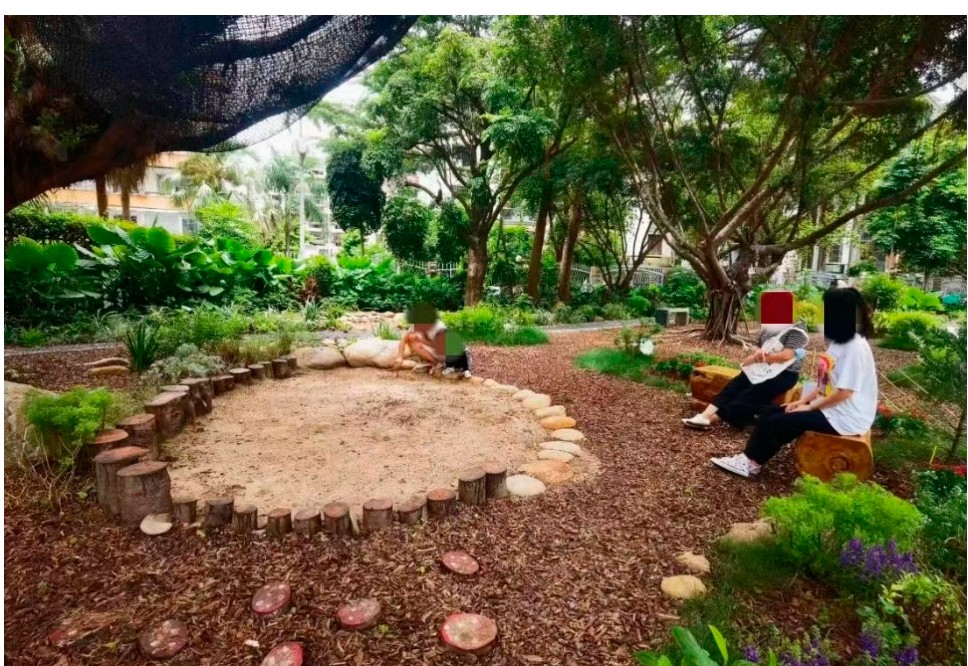

**Figure 7.** *Pen Garden* after transformation. (Source: CHEN Feng, 2020).

*4.4. Nonprofit Organizations Encouraged the Public to Participate in the Governance*

After a two-month construction, a transformed *Pen Garden* was created and entered the maintenance stage. The government's role was to evaluate and assess the effectiveness of the community gardens. After determining that the requirements had been met, the government's continuing contribution at this stage was to provide maintenance funds to the community in the first year to help the community transition to autonomy. More importantly, the nonprofit organizations proposed and adopted a method, in which a leader was elected by the participants collectively to lead the team in maintaining the community garden and finally achieve grassroots autonomy.

The first garden master of *Pen Garden* was elected by the students: a teacher who is also a biology hobbyist. He actively participated in the whole process of garden transformation, organized students from different grades to participate, and used the community garden as his outdoor natural teaching class. This process doubtless helped the teacher and his students establish a relationship with the garden and take the initiative to maintain the garden, such as watering plants, cleaning up leaves, recording plant growth, and performing other daily work (Figure 8).

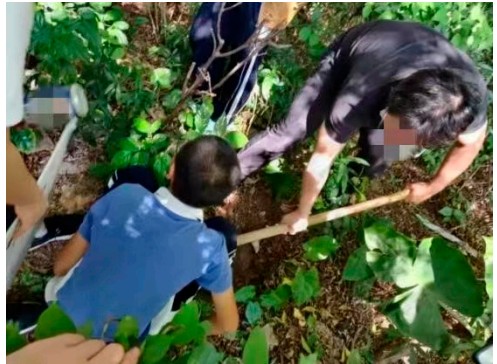 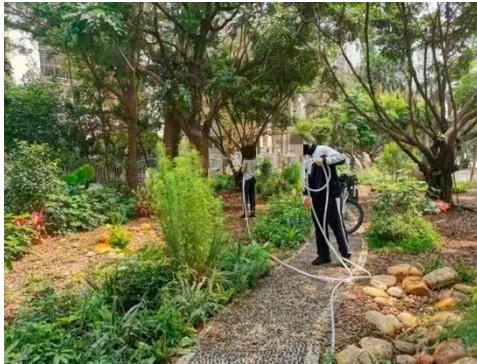

**Figure 8.** Student volunteers carry out garden maintenance under the guidance of teachers. (Source: CHEN Feng, 2020).

*4.5. The Change and Integration of the Participants in the Community Gardens Process*

We interviewed government officials, designers, and students at the *Pen Garden* to obtain their evaluations before and after participation:

(1) Interviewee A from the government said, "*I found that the process of transformation is also the process of youth environmental education. A community garden can become a place for community environmental education.*" This inspiration was also reflected in government policy for the following year. The government consulted experts to develop an environmental education toolkit for gardens provided at no cost to communities to help them carry out daily gardening activities. For many officials who had previously confined their efforts to formulating policies without going on to participate in policy implementation, this action represented a change in their working methods. They are now more willing to go deep into the community, understand people's real needs, and obtain first-hand feedback.

(2) Interviewee E from a designing enterprise said, "*When we studied landscape design at the university, we were reminded to pay attention to the needs of users, but in real projects, we should cater to investors, who always pay attention to business needs.*" Since most of the designers had no experience with the community garden co-construction, the workshop provided a process that changed their work-related thinking. Interviewee E continued, "*At first, I always inadvertently regarded the public at the workshop as investors, but later, I realized it was cooperation. This is different from the traditional way; now, the garden is a design work that meets the needs of users, which makes me look forward to it.*"

(3) The students at the school next to the *Pen Garden* are the primarily participants and the biggest beneficiaries of this project. They were organized to participate from the beginning, then spontaneously carried out garden maintenance. Interviewee I from the school observed, "*Before participating in the construction, I passed this place every day but never paid attention to it. Until the first site survey, I saw many different animals and plants here, and even a bat, which made me feel that this place is alive and full of vitality.*" Meanwhile, Student J from the community school said, "*During the investigation, I found that there were no chairs in the garden, nor in the surrounding areas, which was very unfriendly to the elderly, so I proposed to add chairs in the community garden in the workshop until today I install chairs by myself. When I see people sitting there, it makes me feel very fulfilled.*"

## 5. Discussion

In the "*We Garden*" program, the process of public engagement moved from the top (government) in the form of guidance to the bottom (community) in the form of autonomy. The overarching motivation for launching a community garden laid in the Shenzhen government's desire to change the image of state-led development in urban management [49], reduce financial pressure, and return more green space to the city while encouraging public engagement. Moreover, the project represents a governmental exploration of grassroots autonomy, as well as the implementation of the governance concept of "We are the cities we make" at the national level.

In the first place, in the case of a city that is characterized by its immigrants, starting from the top-down was necessary. Despite Shenzhen's 17.56 million permanent residents, its census-registered population is only 5.84 million [50]. As a result of this difference in the city's population, most of the people living in Shenzhen have no sense of belonging to the community, making their willingness to participate in public affairs relatively low. This environment complicates following a bottom-up governance approach characteristic of Shanghai, Guangzhou, and cities with more native-born residents. Therefore, the *We Garden* scheme, which offered a top-down policy with governmental support and guidance, was of great significance in promoting social participation in public environmental governance in Shenzhen.

However, in light of China's unique socialist institutions, this top-down public participation scheme was itself contradictory. Public attitudes toward the government policy often hover between "for" and "against", and cooperation between the two sides is limited. This circumstance makes the coordination of nonprofit organizations indispensable. On the one hand, nonprofit organizations are government partners. Thus, even as these organizations give full play to the advantages of dealing with public relations as the main promoters, they must also accurately transmit government information in an approachable way that will avoid the public perception of strong governmental intervention in the process of participation. Moreover, this approach provides members of the community with adequate decision-making space, which is the key to effective co-construction. From the community side of the equation, nonprofit organizations serve as community partners, gather feedback about public needs, help members of the public establish an autonomous mechanism, and promote public engagement. In contrast, a scheme that is only promoted by the government without the participation of nonprofit organizations will eventually be dominated by administrative forces and implemented in the form of ordinary garden construction projects, lacking the element of community involvement and shared motivation. Thus, an approach characterized by government investment, recruitment, design, and construction in terms of organizing residents' activities exerts limited influences and is often short-lived, meaning that a government-driven process would make the community garden scheme unsustainable and ineffective.

After the construction of the community garden under consideration had been completed, the government began to encourage grassroots autonomy in the management stage. It is consistent with several cases that transitioned from top-down to bottom-up during the management phase in the literature review [9,10,17,44]. The Shenzhen experiment has improved management feasibility by engaging both top-down and public participation in the planning and construction stages of the community garden, followed by encouraging grassroots autonomy in the management stage, which may be conducive to the sustainability of the community garden as some researchers proposed [31,32]. This kind of grassroots autonomy management deserves more attention in later research, especially support in the form of government funding is later reduced or eliminated.

Through the analysis presented in this paper, we found that the Shenzhen approach differed from the six governance structures summarized in the literature review. The Shenzhen approach is unique compared with two particular structures out of the six. First, in contrast to the top-down with community assistance approach, nonprofit organizations joined as volunteers and served the community free of charge in Shenzhen (some designers as well), which is not paid, as the literature and cases showed [17,29,30,42,43]. Following an approach that includes some of the same elements as the PAS approach and its typical cases [38,45–48], the Shenzhen Municipal Government has taken the initiative to provide funds, land, and professional knowledge to the community on a top-down basis, launching this scheme, while remaining well aware that a solely government-driven scheme would be unsustainable. Therefore, the municipal government has called for the development of more community gardens as a method of public engagement.

These considerations led to formulation of a new (seventh) category: top-down with public engagement driven by nonprofit organizations. We define it as the government taking the lead in launching the policy, providing matching funds and land support, while nonprofit organizations offer coordination and assistance, mitigate administrative intervention, and promote public participation in the planning and design, construction, and management stages of community garden development. The roles and action styles of the main participants are summarized as follows:

(1) Government as the initiator: The community garden scheme has been promoted from top to bottom in the form of policy. The government has been the main supplier of funding and resources for the community. The government monitors the scheme via KPI, promotes and supervises its on-time completion, and assesses the effectiveness of community gardens.

(2)   Nonprofit organizations as the engine: Their participation methods and roles change dynamically through different stages of the project, including government think tanks, project promoters, community partners, and other roles. Their involvement persists throughout the whole process of the project, giving full play to the ability and value in dealing with public relations. These entities also promote effective communication between the government and the public, as well as the public and experts. Nonprofit organizations are the key to the success of the project, and without their participation, enormous obstacles will arise that may be insurmountable.

(3)   Experts/designers and community locals as the public: Experts and designers provide professional knowledge and skills to inexperienced participants. At the same time, this approach also represents a new model for them to build with the community. Unlike traditional landscape projects, instead of magnifying their responsibility, community garden projects call for them to be more like partners cooperating with the community. Community members include residents, nearby workers, and teachers or students, among others. The participation of community members can be characterized a passive-to-active process. In particular, focusing on people's actual needs can foster public engagement. Thus, members of the public are the main part and the biggest beneficiary of social engagement, linking community garden connections to increased social impact and sustainability that extended beyond policy considerations.

## 6. Conclusions

First, the findings supported theorizing a new governance structure (top-down with public engagement driven by nonprofit organizations), which differs from the six previously identified governance structures. Compared with the bottom-up approaches, this governance structure has been shown to facilitate public participation faster and more effectively. Moreover, in comparison to other top-down approaches, this new model may be more sustainable and resilient because it involves more social engagement. We also highlighted the critical role that nonprofit organizations have played throughout the process in fostering the development of community gardens by dealing with public relations and facilitating effective communications among other actors. Nonprofit organizations play an indispensable role, acting as government partners in delivering information from top to bottom, as well as community partners, in helping the public establish an autonomous mechanism. Moreover, these organizations build a communication bridge between the government and the public.

The second point to be made here is historical and evolutionary: community gardens were originally the product of the economic depression, mainly targeted to meet the needs of individuals, but in today's China, this type of project has become a way for the government to promote public participation in urban environmental conservation efforts. From a broad perspective, this governance approach can increase the area of urban green space, uphold standards of high quality, and improve the ecological service function. Meanwhile, from a grassroots perspective, such a program can help public attitudes toward community gardening from passive to active, effectively enhancing public awareness of participation and the decision-making ability available through public participation. While this approach is eminently applicable to the environmental improvement of urban communities, it is also worth referencing in urban projects related to community governance (e.g., urban renewal). Briefly stated, this concept comprises a new sustainable public participation mechanism in urban environmental protection in the Chinese context.

Although the findings reported in this paper provide a new governance structure and dynamics for community gardens in the existing literature, more empirical studies are needed to further test and refine these findings. In addition, since most *We Garden* projects have only recently completed the preliminary management stage, the sustainability of the subsequent management based on grassroots autonomy is a worthy topic for further study.

**Author Contributions:** Conceptualization, X.Z. and Y.Z.; methodology, K.W.; software, D.P.; validation, X.Z., K.W., and D.P.; formal analysis, X.Z.; investigation, D.P.; resources, Y.Z.; data curation, X.Z. and D.P.; writing—original draft preparation, X.Z. and D.P.; writing—review and editing, Y.Z. and K.W.; visualization, K.W.; supervision, Y.Z.; project administration, Y.Z.; funding acquisition, Y.Z. All authors have read and agreed to the published version of the manuscript.

**Funding:** This research was funded by the Marine Special Program of Jiangsu Province in China (JSZRHYKJ202007), the National Natural Science Foundation (U1901215), and the Natural Scientific Foundation of Jiangsu Province (BK20181413).

**Institutional Review Board Statement:** Not applicable.

**Informed Consent Statement:** Not applicable.

**Data Availability Statement:** Not applicable.

**Acknowledgments:** The local yellow books and in situ investigations are highly appreciated. This research was also funded partially by the National Natural Science Foundation (U1901215), the Marine Special Program of Jiangsu Province in China (JSZRHYKJ202007), and the Natural Scientific Foundation of Jiangsu Province (BK20181413).

**Conflicts of Interest:** The authors declare no conflict of interest.

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
