# Peer review of "A New Top-Down Governance Approach to Community Gardens: A Case Study of the “We Garden” Community Experiment in Shenzhen, China"

_urbansci, doi:10.3390/urbansci6020041_

Round 1

Reviewer 1 Report

The authors have an interesting case through which they can explore participation and the co-production/co-construction of urban greenspace. However, the article does not situate itself in the literature and, therefore, does not have a way to use the data collected. As the authors revisit the article, I would urge them to do the following.

  • Completely revise the literature review. Currently, the literature review is a statistical analysis of certain words in the literature. This is wrong. The literature review should examine some of the individual articles and organize them into themes that are important to the story with the goal of finding the theoretical gap that the new case can help fill in. It seems that the authors would benefit from reading more about participation and the co-production of public spaces or public services. Nothing about the current literature review seems to help understand why this case and this research are important.
  • There is no sense of how many interviews were conducted or with whom.
  • Neither the interviews nor the observations seem to be used in the analysis. Indeed, the analysis did not seem to exist
  • The discussion needs to take the data from the analysis and link it back to the literature. It should point to why the research is important.

Author Response

Dear Reviewer,

Thank you for your comments. We have revised and replied to all comments as attached.

Yours sincerely,

Authors

Reviewer 2 Report

The topic of this paper is very interesting. This is a totally pertinent discussion nowadays, taking in account the increase in urbanization, specially in Asia region. The new urban policies, based on the concepts of sustainable and healthy urbanism, have given way to interesting strategies and initiatives to return some "green" to cities.  Community gardens have the advantages not only of contributing to a healthier urbanity but also to a healthier and more sustainable society.

However, there are a number of issues that the authors should take into account:

1- The purpose and objectives of the study are not properly presented in the abstract.

2- In the Literature Review the authors have proven that this is a topic of great interest to the literature. But don't the authors think that what they presented in this section, despite being valid, is short for the dimension and interest of the topic? The main focus areas are presented with regard to community gardens, and the main conclusions of these studies? Bibliographic references are lacking here to frame the topic. There is a lack of a true Literature Review. This section should be improved.

3- In Materials and Methods section, it was difficult to understand if the methodologies had all been applied or not. That conditions the understanding of everything else that is presented later. Interviews with local communities were made specifically to whom? When? How many? The same with the interviews with stakeholders and volunteers. Have these already been done or are they still to be done? Is the samples size representative? The authors should clarify the methodology process to allow replicability and validity.

4- What are the actual results of applying all these methodologies? Where is the Results section? What were the results of the interviews with the various groups?

5- In Discussion section there's a real discussion missing that provides a dialogue between previous research and results. I suggest that this section be improved.

6- The list of references, despite being relevant and current, is too short for an article at the level of this journal.

This is still a very superficial and very descriptive paper, which needs to be worked on. It lacks real content coming from what was applied.

Author Response

(The authors gave the same response as above.)

Reviewer 3 Report

Manuscript ID: urbansci-1549800

Type of manuscript: Article
Title: Community Engagement Model to Support Urban Green Regeneration: An Empirical Research on Joint Construction Community Garden: A case study in Shenzhen, China

Authors: Xunyu Zhang, Yuhong Wang, Kapo Wong, Yuanzhi Zhang

Comments and Suggestions:

The topic and the idea of the study/experiment is interesting, however, the presentation of the study as well as discussion include some weaknesses in my opinion.

1. In general, the use of the term “model” is inappropriate and even incorrect due to the selection of keywords is very limited and does not explain the complexity of the processes and aspects affecting the processes of shaping the community gardens in the context of their planning, design and implementation. This kind of study focused on limited keywords even if found in many literature items (4320) may be accepted as a kind of empirical study, but the scope of aspects that they cover is insufficient. This kind of limited keywords study does not provide enough backgrounds for preparing a “model” in my opinion - it requires a more comprehensive aspects.

I also have the impression that the case study (garden in Shenzhen) even if is related to the mentioned keywords, but shows much more aspects relevant to community garden implementation that need to be addressed. This discrepancy is a weakness of the study and need to be developed in my opinion.

2. The title is too long and too much developed, it should be clear and present the most important idea/aspect of the study in a short and easy way. It must be remodeled according to the content of the manuscript.

3. The Abstract is not well constructed, it should not be divided into two separated “paragraphs”. It contains rather general wordings than specific information/data and/or study results – it must be improved.

4. Key words are related to the topic, but should be developed by adding more terms which are also crucial for presented aspects, e.g. the context of planning which is a topic of Special Issue, etc.

5. Section 1. Introduction - is well prepared and clear, also the research aim is clear. However, to show the aim of the study more clearly, it should be written from a new paragraph (lines 57 to 67).

6. Section 2. Literature review - it is unclear why the Authors have done so much work on studying the use of keywords in many articles (4320), since those selected as most important are not discussed or explained in further description and thus do not show the main trends in shaping community gardens. Indicating the frequency of the leading keywords is the first point for creation of the background for the study, but they require clarification and justification related to their role for the contemporary approach to community garden implementation. This lack of information and relations, and also deficiencies in terms of selected keywords (see pts. 1), is a weakness of this section and must be developed.

7. Section 3. Materials and Methods – this section is not deeply developed. The methods used in the case of Shenzhen are described quite chaotically. Most of them are typical for this type of process, but the Authors did not cite any sources. The Authors also do not use some basic terms in this type of dealing with the local community, managers and other stakeholders, e.g. participatory design to describe more deeply some characteristic processes, etc. – only the term public participation is used as the general one. This shows a bit insufficient knowledge and an incomplete review of literature and methods.

8. Section 4. Case Studies - there is only one case study (not plural form should be used).

The description in this section is much developed, but generally a bit chaotic, and finally there area not listed clearly main important stages of the process related to the creation of Penguin garden, there are no schemes presenting it in a graphic form, etc. Also there is no clear information how long was the process – some information are included in section of Conclusion (lines 290-291) (!). This kind of clear presentation of stages should be used as a part of introduction to the presentation of stakeholders and their participation to the whole process (before subsections 4.1 and 4.2) to understand it better. Another weakness is related to the lack of some general data, e.g. how many representatives of local community and other stakeholders participated to the process, how many students and related to which profession – there are missed information/data which could help to highlight the value of the process and the study itself.

9. Section 5. Discussion – this part is not much developed. Only 3 literature items are cited and thus the discussion sounds more like the Authors interpretation and the scientific soundness may be assessed as insufficient. The aspects discussed are quite selective and have not been compared with other approaches, e.g. in some countries social participation is mandatory in planning and design to ensure community participation in decision-making processes (regarding the description in lines 256-259), etc. Much of the description duplicates generally known information (lines 272-278) but without a comparison to other studies and literature items. The main weakness of the Discussion is that main presented aspects are related to the participatory process itself and engagement of stakeholders, while the aspects of green regeneration as the reason of initiation of presented initiative as well as the positive result of presented cooperation are not developed enough and after all missed, in my opinion.

10. Section 5. Conclusion – the opinions in this section are logical and valuable. However, the preceding sections of the manuscript need to be revised in order to fully follow these conclusions.

11. Others:

- the quality of Figure 2 is low, it is difficult to read presented description

- there are no authors and dates of photos; Figure 7 has wrong number – it should be Figure 5

- punctuation errors - lack of dots, e.g. line 67.

Author Response

(The authors gave the same response as above.)

Round 2

Reviewer 2 Report

Thank you for submitting the revised version of your paper. I appreciate the improvement work that the authors carried out. The improvement is noticeable. About the review version I have some comments and suggestions:

- I still cannot find the purpose of the study clearly stated in the abstract;

- The authors should not have removed Shenzhen from the keywords;

- The research-questions are now clearer. Congratulations on that;

- Literature Review had a huge improvement. In the title capital letters must be removed;

- Point 3 the Authors should keep the designation Materials and Methods. This section is a bit confusing; the information is there but not organized in a coherent way;

- The Results and Discussion section had a great improvement. Congratulations. But, in Discussion could have a more solid connection with literature review.

Author Response

Dear Reviewer,

Thank you for your comments and suggestions. 

We have revised and improved the writing quality with detailed replies as attached.

Yours sincerely,

Yuanzhi Zhang

Reviewer 3 Report

Manuscript ID: urbansci-1549800

Comments and Suggestions:

I appreciate the work of Authors and changes introduced to the manuscript. Regarding the rebuilt version, my comments are as follow.

- the Title - its construction should be more grammar.

- Key words – the name Shenzhen should be not removed, must be kept to pay attention on this specific place of experiment.

- The aim of the study is quite clear, the research questions should be presented in separate lines. However, the next subsections (lines 79-102) sounds more as presentation of the methodology and some parts of methods, than the clear argumentation for conducted experiment. This is too much for this part of the manuscript, some information should be moved to the section 3. Material and Methods.

- The literature review fits to the presented study and create the main background. The cited literature is generally correct, based on not so ‘new’ but still actual information. A weakness is that there are not much relations to other experiments, which could/should be compared in the section of Discussion with this presented in the manuscript.

- The section 3. - in my opinion it would be better to keep the typical form such as: Material and Methods. The form of presentation of this section is still chaotic. The presentation of fieldwork site (case) and methods looks a bit like a mix of the diversity of information. It should be divided into main parts /subsections/ such as the case selection (including its main characteristic), participants characteristics, and then presentation of methods – and they must be named; otherwise the description is rather difficult to read. I suggest to organize subsections such as 3.1, 3.2, 3.3 to make this part easy to understand. As a scientific paper this part must be very clear for readers, and thus should be improved.

- Section 4. Results. The form and order of presentation is correct, and cover wide area of obtained results. However, the introductory part (lines 247-250) has no relation to the Figure 2. It is needed to add sentences introducing the way/order of presentation of results, also the 5 main phases must be listed as an introduction to the further description, etc.

The presentation of results in subsection is is based on using long sentences in some parts, so, it should be more synthetic.

- Discussion – this part of manuscript discusses main aspects presented in the study, however the selection of cited literature is poor. The opinion that “From the perspective of international experience, top-down garden management is difficult to sustain” is very general and not confirmed, especially if Authors cite just one position. There are many experiments presenting both positive and negative results of cooperation of many types of stakeholders and focus on multi-faceted approaches. Another weakness of this part is that the Authors do not relate discussed aspects to the literature presented as the background of the study. It is needed a link to more valuable references, otherwise the relation to the literature review, etc.

- Conclusions – the first subsection (lines 463- 468) are not needed in my opinion.

The conclusion in lines 469-470: “Firstly, the finding theorized a new governance structure (top-down with public engagement driven by non-profit organizations), which differs from the existing six governance structures” is not clear. There are presented in literature experiments that go beyond the conventional approach, but Authors did not develop that contexts in the section of Discussion, and thus this conclusion is rather intuitive than justified.

Regarding some lack of comparison to other experiments in the section of Discussion, the sections of Discussion and Conclusions should be revised / improved.

- English language needs corrections, the construction of some sentences requires general rewording.

Author Response

(The authors gave the same response as above.)

Round 3

Reviewer 3 Report

Manuscript ID: urbansci-1549800

I appreciate all works made by Authors, most of reviewer's suggestions have been introduced. The study presented in the manuscript is valuable and interesting as an experiment . The manuscript can be published in present form.